# Enhancement in Oral Absorption of Ceftriaxone by Highly Functionalized Magnetic Iron Oxide Nanoparticles

**DOI:** 10.3390/pharmaceutics12060492

**Published:** 2020-05-28

**Authors:** Muhammad Kawish, Abdelbary Elhissi, Tooba Jabri, Kanwal Muhammad Iqbal, Hina Zahid, Muhammad Raza Shah

**Affiliations:** 1International Center for Chemical and Biological Sciences, H.E.J. Research Institute of Chemistry, University of Karachi, Karachi 75270, Pakistan or kawishiqbal02@gmail.com (M.K.); or toobaasif137@gmail.com (T.J.); or kanwalmuhammadiqbal43@gmail.com (K.M.I.); 2College of Pharmacy, QU Health, and Office of VP for Research and Graduate Studies, Qatar University, Doha 2713, Qatar; aelhissi@qu.edu.qa; 3Faculty of Pharmaceutical Sciences Dow University of Health Sciences Karachi, Karachi 74200, Pakistan; hina.zahid@duhs.edu.pk or

**Keywords:** iron oxide nanoparticles, surface functionalization, biocompatibility, ceftriaxone, oral delivery

## Abstract

The present study aims at the development, characterization, biocompatibility investigation and oral bioavailability evaluation of ceftriaxone (CFT)-loaded N′-methacryloylisonicotinohydrazide (MIH)-functionalized magnetic nanoparticles (CFT-MIH-MNPs). Atomic force microscopy (AFM) and dynamic light scattering (DLS) showed that the developed CFT loaded MIH-MNPs are spherical, with a measured hydrodynamic size of 184.0 ± 2.7 nm and negative zeta potential values (–20.2 ± 0.4 mV). Fourier transformed infrared spectroscopic (FTIR) analysis revealed interactions between the nanocarrier and the drug. Nanoparticles showed high drug entrapment efficiency (EE) of 79.4% ±1.5%, and the drug was released gradually in vitro and showed prolonged in vitro stability using simulated gastrointestinal tract (GIT) fluids. The formulations were found to be highly biocompatible (up to 100 µg/mL) and hemocompatible (up to 1.0 mg/mL). Using an albino rabbit model, the formulation showed a significant enhancement in drug plasma concentration up to 14.4 ± 1.8 µg/mL in comparison with its control (2.0 ± 0.6 µg/mL). Overall, the developed CFT-MIH-MNPs formulation was promising for provision of high drug entrapment, gradual drug release and suitability for enhancing the oral delivery of CFT.

## 1. Introduction

Oral drug administration is the most established strategy for achieving improved clinical outcomes of drugs. The oral route is devoid of discomforts associated with the intravenous route, thus providing higher patient compliance and no need for hospital admission to administer the dose. The oral route can also be used for delivery of less water-soluble drug molecules that cannot be administered through other routes [1]. However, drugs delivered through the oral route have to face various obstacles such as instability and insolubility in the harsh acidic gastric environment, degradative effects of digestive enzymes and poor absorption through the intestinal epithelium which results in poor drug bioavailability [2,3]. Recent developments in the field of nano-drug delivery systems have demonstrated improved oral delivery of various drug formulations through protection from enzymatic degradation and improved absorption through the intestinal epithelium. They also improve the oral delivery of drugs through organ-specific localization with a controlled release manner [4].

Magnetic nanoparticles (MNPs) are widely considered a promising drug delivery system due to their distinct advantages such as well-documented bio-safety, ease of preparation and handling, the possibility for controlling their characteristics, affordability of materials needed and the possibility of targeting the drugs to a desired location. Furthermore, core-shell MNPs have attracted much attention due to their multifunctional properties such as small size, superparamagnetism and low toxicity [5,6]. Silica-coated MNPs are one of the most extensively used NPs which possess very high specific surface with abundant Si–OH or Si–NH_2_ groups with the ability to react with proper functional groups [7,8]. Besides these advantages, various issues such as their rapid aggregation and early clearance from blood circulation by the reticuloendothelial system (RES), reduce their therapeutic efficacy [8,9]. Encapsulation of MNPs within biodegradable and biocompatible substances is one approach for addressing these issues [10].

Additionally, encapsulation of MNPs with other molecules can be used by inserting desirable and diverse functionality for the conjugation of drugs [11]. Surface functionalization of MNPs for specific purposes such as enhancement in the oral bioavailability of drugs has attracted increased attention. MNPs bearing a surface enriched with hydroxyl groups can constitute a potential mucoadhesive carrier for enhancing oral bioavailability of their loaded drugs [12].

Ceftriaxone (CFT), a third-generation cephalosporin, is capable of inhibiting the biosynthesis of the bacterial cell wall. CFT is widely prescribed for the treatment of various infections such as urinary tract infections, endocarditis, meningitis and pneumonia. However, oral delivery of CFT is a real challenge, constituting one of the limitations of this drug. It belongs to class III drugs in the biopharmaceutical classification system and suffers from lower membrane permeability. This leads to poor biological performance of the drug after its oral administration [13,14]. CFT is poorly absorbed through the mucosal membrane of the intestine due to its two negatively ionized carboxyl groups [15]. Thus, a versatile nanocarrier surface functionalized with molecules having multi-hydroxyl groups is highly desirable for oral delivery of CFT.

For the first time, the present investigation was based on the delivery of CFT through drug conjugated nanoparticles (NPs) for the enhancement of oral bioavailability of the drug. For this purpose, isoniazid was functionalized on MNPs to get a supramolecular core-shell nanostructure. We investigated whether the developed NPs were capable of stacking and stabilizing CFT electrostatically with the aid of pyridine moiety in isoniazid against harsh stomach conditions. We further evaluated the ability of NPs to offer therapeutic loadings of CFT and the mechanism of interaction between CFT and the NPs. The synthesized nanoparticles were also characterized in terms of their drug loading capacity, size, surface charge and drug release profile. In addition, biocompatibility and stability studies were conducted and using a rabbit animal model, the in vivo oral bioavailability investigations were performed.

## 2. Methods and Materials

### 2.1. Materials

All solvents utilized in experiments were of HPLC grade and obtained from Fisher Scientific, (Loughborough, UK) Ferric sulphatehexahydrate Fe_2_(SO_4_)_3_.6H_2_O, ferrous sulphate heptahydrate (FeSO_4_.7H_2_O), 3-(trimethoxysilyl)propyl methacrylate (MPTES), 4-dimethyl aminopyridine (DMAP), azobisisobutyronitrile (AIBN), ammonium hydroxide, dicyclohexylcarbodiimide (DCC) and methacrylic acid (MMA) were obtained from Sigma-Aldrich (St. Louis, MO, USA), Merck (Darmstadt, Germany and ceftriaxone (intravenous-Infusion) and isoniazid through Getz Pharmaceuticals (Karachi, Pakistan). Fresh human plasma was obtained from Ziauddin University Hospital (Karachi, Pakistan).

### 2.2. Synthesis of MIH

MMA (1.88 g, 21.8 mmol), DMAP (0.26 g, 2.1mmol) and DCC (4.51 g, 21.8 mmol) were dissolved in tetrahydrofuran (THF) (40 mL) in a round bottom flask connected with a condenser. The reaction was stirred for 10 min at 60 °C under Ar atmosphere. Isoniazid (1.0 g, 7.29 mmol) was added later, and the reaction was refluxed for 24 h followed by monitoring *via* thin-layer chromatography using DCM and MeOH (9:1 *v*/*v*) as a solvent system. The resulting mixture was concentrated in vacuo and the concentrate was then subjected to column chromatography using flash silica as a stationary phase. Desired pure compound was obtained using hexane and ethyl acetate (6:4 *v*/*v*) as a mobile phase.

Rf, 0.56 (DCM : MeOH, 9:1, *v*/*v*),yield 50%; M.P., 99.3–100.4 °C, EI-MS 205.1, ^1^H NMR (300 MHz MeOD); δ ppm, 8.70 (d 2Hpyridine), 7.85 (d 2H pyridine), 5.86 (s 1H amide), 5.51 (s 1H amide), 3.02 (s 3H CH_3_).

### 2.3. Preparation of MIH-MNPs and CFT-MIH-MNPs

First, narrow range MNPs were prepared by precipitating Fe(III) and Fe(II) in an alkaline medium by a co-precipitation technique [16]. Then, synthesized MNPs underwent surface coating with MPTES through silanization reaction [17] in such a manner that the ratio between MNPs and MPTES remained 1/6 upon incubation for 4 h at 60 °C. MIH coated MNPs was obtained through radical polymerization method. Briefly, a solution of MPTES-modified MNPs (0.15 g) and MIH (0.950 g, 4.63 mmol) in a ratio of 1:6 werestirred for 1 h at 60 °C to facilitate the pre-polymerization process. After stirring for 10 min, AIBN (1.3 g, 7.91 mmol) was added to the resulting mixture and was refluxed for 24 h under Ar atmosphere at 60 °C. The obtained MIH-MNPs were washed sequentially with acetonitrile (ACN) and dried at –20 °C on a freeze dryer (Vritis 25 S.R.C., New York, NY, USA.) overnight. CFT loading was performed with slight modification of a previously published protocol [18]. Briefly, nanosuspension of MIH-MNPs (1mg/mL) was introduced in various concentrations of CFT (1–5 mg/mL) insuch a way that the ratio between the nanoparticles (NPs) and CFT remained 1:1 (*w*/*w*). This was then incubated on a rotary shaker at 200 rpm for 24 h to facilitate drug uptake. The resultant CFT-MIH-MNPs were removed via centrifugation at 15,483 G, and the supernatant containing the unloaded drug was quantified on a UV-Vis spectrophotometer (Shimadzu 1800 series) Shimartzu, Kyoto, Japan. The blend containing a higher amount of drug entrapment was considered for further analysis.

### 2.4. Particle Size, Size Distribution, Zeta Potential and Morphology Studies

The average hydrodynamic diameter and PDI of vacant and CFT loaded MIH-MNPs was investigated via the Zetasizer (Zetasizer Nano ZS90 Malvern Instruments, Malvern, UK). Concisely, diluted NPs were transferred to a transparent plastic cuvette with caution to avoid bubble formation. The cuvette was then placed in the cell holder of the instrument and analysis was taken at 90 degree scattering at 25 °C. The medium viscosity and refractive index options in the instrument’s software were kept constant at 1.33 and 80.4 mPa, respectively. NPs were also characterized for morphology using AFM (AFM, Agilent 5500) Agilent, CA, USA. A drop of the formulation was placed on a mica slide and air dried at ambient temperature and placed on a microscope. The morphology was investigated at non-contact mode.

### 2.5. FTIR Spectroscopy

FT-IR spectra were obtained by Shimadzu IR-470 spectrometer (Shimadzu, Kyoto, Japan) having a resolution of 4 cm^−1^, to elucidate the self-assembled layer formed on the surface of NPs. A small number of powdered NPs was mixed with KBr and subjected to a pressure value of 200 Psi to obtain self-supporting disks.

### 2.6. Stability of NPs in Simulated Gastric Fluids (SGF)

The persistence of the MIH-MNPs mentioned above, and CFT-MIH-MNPs in SGF was evaluated with a slight amendment of previously published studies [19]. Briefly, the stock SGF was prepared without pepsin by mixing 300 mM NaCl with a solution containing 840 mM HCl (pH 1.2). Pancreatin excluding intestinal fluid was prepared and composed of 800 mM NaHCO_3_ (pH 7.4). At the predetermined interval, the sample was taken for determination of size and PDI by DLS at 37 °C using the Zetasizer Instrument (Malvern Instruments Ltd., Malvern, UK). UV-V is spectroscopy analysis was also performed to evaluate the stability of CFT in the SGF medium.

### 2.7. Development of HPLC Protocol for the Determination of CFT in Blood Plasma

The availability of CFT and internal standard cephalexin (CEP) in plasma was determined through reverse phase HPLC protocol developed and validated in our laboratories. The protein precipitation method was taken into account for extracting drug from plasma. Plasma (200 µL) containing both drugs was blended and vortexed with ACN (400 µL) for 2 min and the precipitate was separated by centrifugation at 15,483 G for 10 min. The acquired supernatant was concentrated in vacuum and reconstituted in buffer (TBAH, 100 µL; pH 7.0). A volume of 25 µL was injected into the HPLC system Shimadzu, LC20A (Shimadzu, Kyoto, Japan) equipped with a purospher C18 (5 µm, 250 × 4.6 mm) column. The mobile phase was comprised of ACN and TBAH buffer (0.005 M, pH 7.0) in a ratio of 25:75. The flow rate was adjusted to 1 mL/min, and the wavelength selected was 220 nm.

### 2.8. Entrapment Efficiency Determination

The targeted MIH-MNPs were exploited for drug entrapment efficiency, according to previously reported protocol [20]. The obtained supernatant after centrifugation of CFT-MIH-MNPs suspension was subjected to successive dilution and analyzed at 241 nm on a UV-Vis spectrophotometer. The drug entrapment efficiency (EE) was calculated by using the following equation:EE (%) = [(A _initial_ − A_free_)/A_initial_] ×100, 
where A_initial_ is theamount of CFT taken for loading and A_free_ is theamount of unloaded CFT.

### 2.9. In Vitro Release Study

The drug release experiment was performed in triplicate using three different batches, and the samples were drawn at certain time intervals from the release medium with the replacement of fresh buffer of respective pH. The amount of drug released was determined by spectrophotometry at 241 nm, while maintaining the temperature at 37 °C. Samples (5 mg equivalent CFT loaded MNPs were dispersed in buffer, 4 mL; pH 4.0 or pH 7.0) and immersed in a 50 mL release medium (pH 4.0 or 7.0). The release rate studies were performed using the dialysis membrane of 12,000 kDa using the dialysis method described in a previous study [21].

### 2.10. Hemocompatibility Study

Ethylenediaminetetraacetic acid (EDTA) stabilized human blood samples (5 mL) were added to 10 mL of PBS. Red blood cells (RBCs) were then isolated from serum by centrifugation at 3871 G for 10 min. The RBCs were further washed five times with 10 mL of PBS solution. Purified RBCs were diluted to 50 mL of PBS. Prior to nanoparticle exposure, the absorbance spectrum was analyzed. Herein, RBCs incubation with Triton X was used as the positive test. Then, 0.2 mL of diluted RBC suspension was added to 0.8 mL of MIH-MNPs solution in 200–1000 μg/mL concentrations, followed by vortex mixing. All sample tubes were kept in static condition at room temperature for 3 h. Finally, the mixtures were centrifuged at 15,483 G for 10 min, and 1.5 mL of supernatant of each sample was transferred to a cuvette. The absorbance values of the supernatants at 540 nm were determined by UV-VIS spectrophotometer. The percent hemolytic activity of RBCs was calculated using the following equation:% H.A. = Rs/Rc ×100.

Rs: absorbance of sample; Rc: absorbance of positive control; % H.A.: hemolytic activity in percent.

### 2.11. In Vitro Cytotoxicity Study

The synthesized nanocarrier was investigated for its cytotoxicity using MTT assay. Both 3T3/NIH and HeLa cells were refined in Dulbecco’s Modified Eagle Medium (DMEM) having fetal bovine serum (10%) and antibiotics (streptomycin and penicillin, about 50 U/mL) at room temperature and under carbon dioxide (5%) humid atmosphere. Both cell lines were individually incubated into 96-well plates with 8 × 10^3^ and 6 × 10^4^ cells/well thickness in 200 µL refined media. After incubation for 24 h, fresh media (200 µL) consisting of NPs was introduced at various concentrations for 100 μg/mL. Incubated cells in media without NPs were used as a negative control and developed for 48 h. 3-(4,5-Dimethylthiazol-2-Yl)-2,5-Diphenyltetrazolium bromide solution (MTT) in PBS was introduced in each well (20 µL; 5 mg/mL). The unreacted portion was expelled after 4 h of incubation. The resulting formazan crystals were dissolved and introduced at 200 μL DMSO per well and analyzed at 570 nm in a microplate reader. For positive control and reference standard, doxorubicin (DOX) and cyclohaxanamide (CHX) were used. The following equation was utilized to determine % cell viability: Cell viability % = At/Ac× 100.

At: mean of absorbance value of test sample; Ac: mean of absorbance value of a control.

### 2.12. In Vivo Oral Pharmacokinetics Studies

Animal studies were conducted under institutional guidelines by following the principals of institutional animal care. The studies were approved by the Ethics Committee of International Center for Chemical and Biological Sciences, University of Karachi (Project identification code ICCBS/CBSCR/IEC-014/2018; approved date 06-04-2018). All animals were healthy and purchased for Dow University of Health Sciences OJHA campus Karachi, Pakistan. Rabbits (*Oryctolaguscuniculus*) with 1.2 kg normal body weight were chosen for the assessment of oral in vivo pharmacokinetics studies. Rabbits were held under standard quarantine conditions with free access to food and water for 12 h. Before conducting the in vivo studies, rabbits were kept under fasting conditions but with free access to water. Rabbits were divided into three distinct categories (*n* = 4). The first category was given CFT-MIH-MNPs suspension at 25 mg/kg body weight portion. Rabbits of the second and third categories were provided with CFT in oral aqueous solution at 25 and 50 mg/kg body weight. Blood samples (1mL) were collected via marginal ear vein using an insulin plastic syringe at various intervals within 24 h. After centrifugation at 2688 G for 10 min, plasma was acquired and stored at –20 °C. Pharmacokinetics parameters were assessed by linear trapezoidal standard using a non-compartmental model from individual drug plasma concentrations with reference to a time curve for CFT after oral administration of nanoformulation and controls separately. The estimations of maximum drug plasma concentration (C_max_) and time to reach maximum plasma concentration (T_max_) were obtained from the drug plasma concentration curves. These parameters were utilized to ascertain the pharmacokinetics parameters area under the curve from zero to last quantifiable value (AUC_0-24_), mean residence time (MRT) and clearance (Cl).

### 2.13. Statistical Analysis

The investigations were performed with sample size of *n* = 3, and outcomes were depicted as mean ±SEM. Two-way ANOVA pursued by Beffroni test was applied for the examination of different variables, and Student’s *t*-tests were utilized for correlation of two categories. *P* values less than or equal to 0.05 indicated significant differences between the compared groups.

## 3. Results and Discussion

### 3.1. Synthesis of *MIH*

In order to develop MIH, MMA was linked with isoniazid through carbodiimide-triggered coupling reaction (Figure 1) which is well known for its reproducibility, selectivity and high yield as previously reported [22,23]. DMAP was added in catalytic amount owing to its multifunctional characteristics as a base and in intermediate formulations by knocking out urea. The whole process was performed under Ar atmosphere to avoid atmospheric interference [24].

The synthesized MIH shows EI-MS spectrum has M^+^ at *m/z* 205 which coincides with the theoretical weight of the compound having a formula containing odd nitrogen C_10_H_11_N_3_O_2_. The ^1^H NMR spectra of synthesized MIH compound shows a series of aromatic doublet around δ 7.8 ppm and 8.7 ppm of 2H. Two olefinic protons have also appeared as singlets around δ 5.5 ppm and 5.8 ppm of 1H. A singlet observed around δ 3.02 ppm corresponds to the methyl group. The reaction yielded 50% as a product.

### 3.2. Preparation of MIH-MNPs and CFT-MIH-MNPs

Among different routes for the synthesis of MNPs, the co-precipitation method is most preferred for the preparation of MNPs [25]. This method is economical, convenient and provides good control over size distribution and morphology under ambient conditions [26]. Uncoated NPs tend to aggregate and oxidize; therefore, it is necessary to functionalize their surface to prevent aggregation [25]. MPTES was added to stabilize and functionalize pi-bonds onto the surface of MNPs to initiate polymerization [27]. Finally reflux-polymerization was performed using MIH as a monomer and AIBN as an initiator.

The synthetic scheme of silane-coated MNPs and its MIH conjugated nanostructures is reported earlier [28] and depicted in Figure 1. FT-IR spectra of iron oxide nanoparticles (IONPs) and its surface-functionalized analogues is shown in Figure 2. The synthesized IONPs show characteristic absorption around 638.4 cm^−1^ which is attributed to vibration of Fe–O bonds. Furthermore, absorptions at 1641 cm^−1^ and 3442.83 cm^−1^ correspond to stretching and bending vibrations of the hydroxyl groups on the surface of NPs. The functionalization of (MPTES) induces characteristic absorption around 1656 cm^−1^ and 1775.42 cm^−1^ of *α*, *β* unsaturated bond and ester (C=O), respectively [29]. A frequency of 1056.96 cm^−1^ observed in the finger print region corresponds to C–O bending. Fabrication of MIH on silane-coated NPs gives a pair of frequencies at 1745.62 cm^−1^ and 1690 cm^−1^, and corresponds to the (C=O) of ester and amide, respectively. A stretch around 1445.6–1560.26 cm^−1^ corresponds to aromatic ring stretching. The bending vibration around 3463.08 cm^−1^and 1641 cm^−1^ corresponds to N–H. The characteristic absorption gives evidence for successful functionalization of MIH molecules on the surface of NPs.

To investigate the drug entrapment, FT-IR analysis was conducted of the CFT and CFT-MIH-MNPs loaded formulation [30]. The CFT molecule shows a characteristic absorption at 1748 cm^−1^ and 1554 cm^−1^ corresponding to *β* lactam C=O and C=N, respectively [31]. The band shifted to 41 cm^−1^ of C=O and 7 cm^−1^ of C=N, lower wavenumbers indicating that *β* lactam moiety was involved in coordination with MNPs. Moreover, the asymmetric stretch of OH and NH_2_ would appear at 3455.3 cm^−1^ and 3288.4 cm^−1^ (Figure 3), respectively. The band of OH was shifted to 3485 cm^−1^ and the peak of NH_2_ was eliminated, showing that both groups are involved in coordination with nanostructures (Figure 3). The stretching frequency of COO^-^ appeared at 1600 cm^−1^ as illustrated in Figure 3. The peak was shifted to 1567 cm^−1^ when it was loaded onto nanostructures, indicating conjugation through that group [32]. It is evident from FTIR comparison that CFT was adsorbed through chelation on MIH-MNPs via amine, carboxylic, beta-lactam and hydroxyl groups.

### 3.3. Particle Size, Size Distribution, Zeta Potential and Morphology Studies

Dynamic light scattering (DLS) technique showed an average hydrodynamic diameter of 168.0 ± 7.3 nm and 184.0 ± 2.7 nm for MIH-MNPs and CFT-MIH-MNPs, respectively. Thus, the incorporation of CFT onto the surface of NPs increased the size of the MNPs. Zeta potential of MNPs was found to be –11.4 ±1.32 mV, and upon fabrication with an MIH molecule, the zeta potential intensity increased to –17.7 ± 0.40 mV, indicating successful conjugation of MIH onto the surface of MNPs. When CFT was introduced, the zeta potential further increased to –20.2 ± 0. mV (Table 1), indicating binding of CFT onto the surface of MNPs. The studies were performed at neutral pH (pH = 7.4). Higher zeta potential may indicate better colloidal stability which was the case when the drug was hosted by MIH-MNPs. This assumption was validated further under various conditions in subsequent studies within this report. The polydispersity index (PDI) describes the uniform dispersion of colloidal suspension. PDI values higher than 0.3 indicate polydispersity of the particles [33]. In our study, the PDI of drug-free and CFT-loaded MIH-MNPs were found to be 0.237 and 0.265 (Table 1), respectively, indicating NPs homogeneity. AFM images showed a nearly spherical shape of the MIH-MNPs, which became slightly distorted after drug incorporation, possibly due to filling of hollow matrices within the NPs (Figure 4). Moreover, an AFM image showed uniform size distribution of drug-loaded NPs, supporting PDI findings (Table 1).

### 3.4. Stability in Simulated Gastric Fluids (SGF)

Targeting intestinal delivery requires stability in a gastric environment. IONPs are greatly influenced by ionic strength and pH of the medium [34]. Synthetic gastrointestinal environment was developed in vitro to evaluate the stability of MIH-MNPs and CFT-MIH-MNPs for 72 h. The outcome of the study showed prominent stability of MIH-MNPs and CFT-MIH-MNPs under a gastric environment (i.e., size and PDI remained under 15% variation), which was due to the inherent high zeta potential and surface functionalized pyridine moiety. This tends to protonate at a low pH and give rise to electrostatic repulsion between the NPs, which contributes to stabilizing the particles under the simulated gastric environment (Figure 5A,B). Intestinal stability was evaluated by incubating MNPs in synthetic intestinal fluid, prepared without enzymes and in lysosomal fluid. Particle size and PDI of MIH-MNPs and CFT-MIH-MNPs remained unchanged, indicating nanoparticle stability in the intestinal environment (Figure 5C,D).

### 3.5. Development of HPLC Protocol for the Determination of CFT in Blood Plasma

In order to investigate the concentration of CFT in blood after administration of CFT-MIH-MNPs in vivo, the HPLC method was developed and validated for the determination of CFT in blood plasma. The optimized procedure indicated stability within three freeze-thaw cycles owing to acceptable variations in results (from 90.6% to 110%). The retention time of CFT and CEP was found to be 13.5 and 8.5 min, respectively. The developed method showed an excellent linear relationship with an R^2^ value of 0.998 over a concentration range of 1–50 µg/mL of CFT in blood plasma. The method was validated as inaccuracy (% CV = 2.60%) and precision (% CV = 1.9 %) were well within compliance limits.

### 3.6. Drug Entrapment Efficiency

One of the most significant properties of a nanocarrier is its ability to act as a delivery system by accommodating the drug and transporting it to the desired site/organ, eventually being able to release the drug over a prolonged period. CFT consists of weak acidic molecules with a pKa of 3, 3.2 and 4.1 and an early neutral pH of about 6.7 [35]. The entrapment efficiency of CFT-MIH-MNPs was found to be 79.4% ± 1.5% in comparison to 73.13% ± 2.4 % of drug-loaded MPTES-MNPs. The slight increase in drug loading may be attributed to increased secondary interactions in terms of pi-pi stacking, and increased hydrogen bonding offered by a coated MIH molecule (Figure 6). As shown in Table 1, the surface charge of drug-free MIH-MNPs was negative at a neutral pH medium. Thus, the secondary electrostatic interaction and chelation between MIH-MP and CFT could be the mechanism of drug entrapment [36]. Increment in zeta potential demonstrated the effective drug hosting as the drug binds to NPs.The negatively charged core tends to interact more strongly with the drug, owing to its lowered pKa, which may make MIH-MNPs a useful supramolecular hosting system for provision of significant entrapment of CFT. CFT entrapment values found in our study are higher than entrapment reported using other nanocarriers such as chitosan NPs in which the proportion of drug entrapped was dependent on drug concentration (EE up to50%) [30].

### 3.7. In Vitro Drug Release Study

The release profile of CFT molecules from MIH-MNPs analogue was estimated at a physiological pH of 7.4 and a weakly acidic environment of pH 4.0, under maintained physiological temperature (Figure 7). Interestingly, CFT-MIH-MNPs show a higher release of 24.3% ± 1.5% at pH 7.4 in comparison with pH 4.0, being 18.89% ± 0.3% at 6 h. The lower drug release at an acidic pH may be attributed to protonation of the pyridinium nitrogen, which promotes the electrostatic interactions between CFT and NPs. The higher release at physiological pH (i.e., pH 7.4) may support our hypothesis of stabilization of CFT molecules in acidic pH [37].

### 3.8. Hemocompatibility

The interaction of surface-functionalized magnetic NPs with negatively charged membranes was studied via the hemolysis test [38]. The hemolytic effects of MIH-MNPs were performed at five different concentrations (200, 400, 600, 800 and 1000 µg/mL) against triton X as a positive control, and the released hemoglobin was quantitatively analyzed at 541 nm (Figure 8a). When the concentration was 1 mg/mL, the hemolytic activity was less than 10% which suggests blood compatibility findings [37,39] and encouraged us to proceed with subsequent in vitro and in vivo studies.

### 3.9. In Vitro Cytotoxicity

Although MNPs are well reported for biomedical applications, it is essential to perform cytotoxic analysis to evaluate safety of the designed MNPs [25]. The mitochondrial deterioration within the cellular core after uptake of MNPs was studied using the MTT method. The cytotoxicity of designed MNPs was evaluated in a dose-dependent manner (Figure 8b), and shows the findings of cell viability of MIH-MNPs against 3T3 and Hela cell lines after 48 h. Results show that 3T3 cell lines demonstrate a dose-dependent biocompatibility. Cell viability at 100 μg/mL was found to be 73.0% ± 1.2%, which in comparison, shows higher viability than what was previously reported for MNPs [40]. The decrease in cellular viability of 3T3 may be attributed to the production of reactive oxygen species (ROS) upon MIH-MNPs uptake [41]. However, higher biocompatibility was observed in the case of Hela cell lines, which was found to be 98.0% ± 0.2%, respectively. Biocompatibility within less than 100 μg/mL encourages exploring MIH-MNPs for in vivo applications.

### 3.10. In Vivo Oral Pharmacokinetic Studies

CFT plasma concentrations curve at various time intervals after oral dosing in the form of its solution (50 mg/kg) and CFT-MIH-MNPs formulation (25mg/kg) are shown in Figure 9. Pharmacokinetics parameters were evaluated from plasma concentrations versus time plot and are detailed in Table 2. The maximum plasma concentration (C_max_) of 14.4 ± 1.8 µg/mL was achieved for 25 mg/kg dose at 6 h when CFT was entrapped in MIH-MNPs, with reference to its control which was found to be 2.0 ± 0.6 µg/mL for a 50 mg/kg dose at 1.5 h. This was in agreement with previously published studies [42]. In our in vivo experiments, no drug was detected in HPLC chromatograms when the control CFT was administered at 25 mg/kg. This is attributed to the acid-labile property and low penetration nature of CFT when given in a traditional solution form [43]. This provided evidence that our MNPs formulation protected CFT from degradation in the acidic environment of the stomach owing to electrostatic interaction between protonated pyridine moiety and negative CFT molecule, which is further supported by the fact that CFT-MIH-MNPs formulation was stable in the simulated gastric fluid, as shown earlier in Section 3.4.

Interestingly, the minimal dose of 25 mg/kg in the form of CFT-MIH-MNPs enhanced the C_max_ by 7.2 fold and T_max_ to 6 h. This is significantly higher with reference to its control, which at the same dosage showed no detection of CFT in the blood plasma. Moreover, the CFT-MIH-MNPs formulation showed in vivo resistance represented by a mean residence time (MRT) of 12.2 ± 0.2 h and decreased clearance of 1.6 ± 0.2 L/h, in comparison to the control which had values of 8.4 ± 0.1 h and 3.91 ± 0.4L/h, respectively. This shows the stability and sustainability of CFT-MIH-MNPs formulation caused by the significant increase in drug plasma concentration arising from the higher absorption of the drug through GIT [44].

The nano size and negative surface charge also enhances the lymphatic drainage across the intestine [45]. Furthermore, the drug entrapped in magnetic NPs can prevent its degradation in the harsh acidic environment [46,47]. Nanoformulation may help by passing the first-pass hepatic metabolism and enterohepatic circulation by passing through portal circulation [48], thereby enhancing oral absorption. Compounds rich in OH and NH_2_ functional moieties may possess bioadhesive properties [49]. Bio adhesive functional moieties, stability in a harsh gastric environment, minimization of first-pass metabolism and negative surface charge of the NPs are all potential contributors to the enhancement of oral absorption of CFT across the GIT epithelium.

Finally, reflecting on the procedure used to prepare CFT-MIH-MNPs, another antibiotic (isoniazid) was employed to stabilize CFT. It is well known that antibiotic drug combinations may elicit additional or synergistic therapeutic effects but may also result in a higher potential of antibiotic resistance. An enhanced antibacterial effect has been reported by combining cefotaxime (another third generation cephalosporin) with isoniazid and rifampicin [50]. Future investigations may explore the potential benefit or harm of using isoniazid in the chemical composition of the present MNP formulation in vitro and in vivo. Furthermore, our research group will consider conducting studies to explore more directions related to the use of MNPs, such as the role of MNPs as acoustic imaging tools [51].

## 4. Conclusions

We have designed a biocompatible MIH conjugated IONPs to enhance stability and permeability of CFT across the GIT epithelium. The NPs were found to be highly biocompatible, showing good GIT persistence and the ability to entrap therapeutic proportions of the drug. The developed MIH-MNPs formulation considerably enhanced the in vivo oral bioavailability of CFT with reference to its control using an animal model. This study suggests that the resulting nano range carrier can be a promising strategy for enhancing the oral bioavailability of low penetrating and unstable drugs across the GIT epithelium, taking the CFT model as a successful example.

## Figures and Tables

**Figure 1 pharmaceutics-12-00492-f001:**
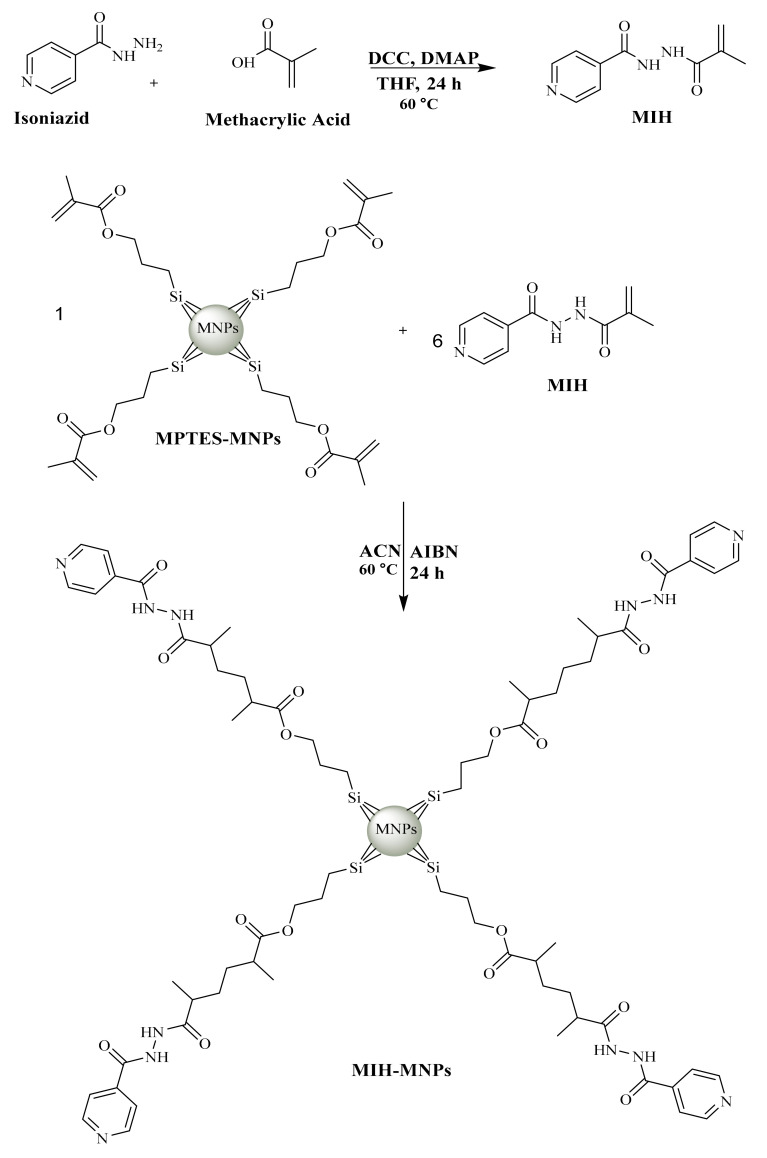
Synthetic scheme of MIH and MIH coated magnetic nanoparticles (MIH-MNPs) [28].

**Figure 2 pharmaceutics-12-00492-f002:**
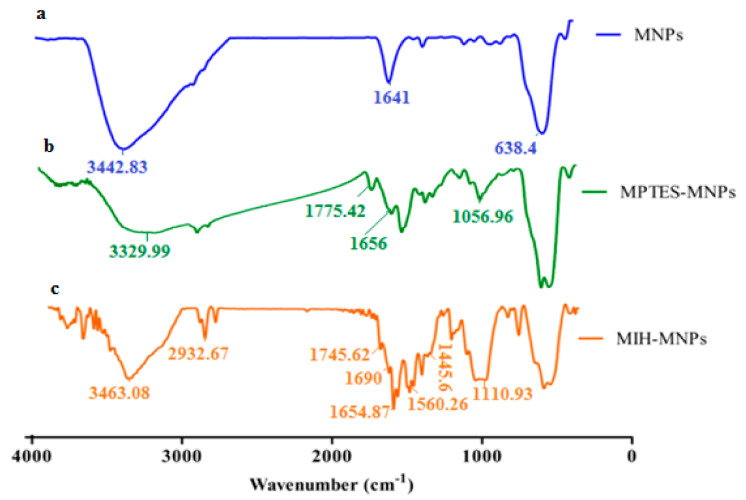
(**a**) Fourier transformed infrared spectroscopic (FTIR) spectra of magnetite nano particle, (**b**) spectra of silane coated nano particle, and (**c**) spectra of synthesized MIH-MNPs.

**Figure 3 pharmaceutics-12-00492-f003:**
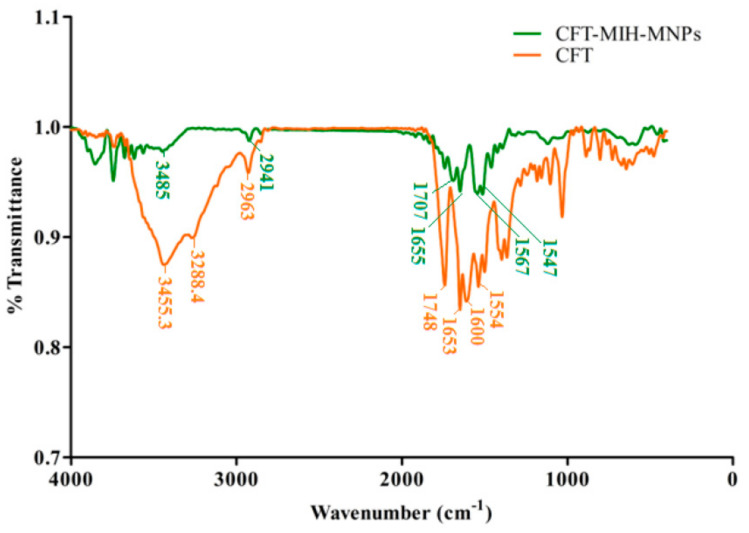
FTIR spectra of drug-loaded CFT-MIH-MNPs along with ceftriaxone.

**Figure 4 pharmaceutics-12-00492-f004:**
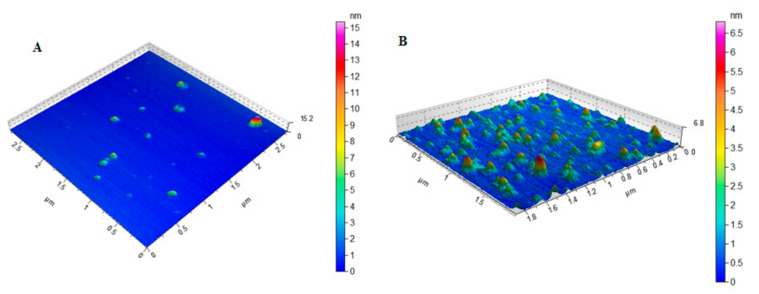
AFM images of (**A**) MIH-MNPs and(**B**) CFT-MIH-MNPs.

**Figure 5 pharmaceutics-12-00492-f005:**
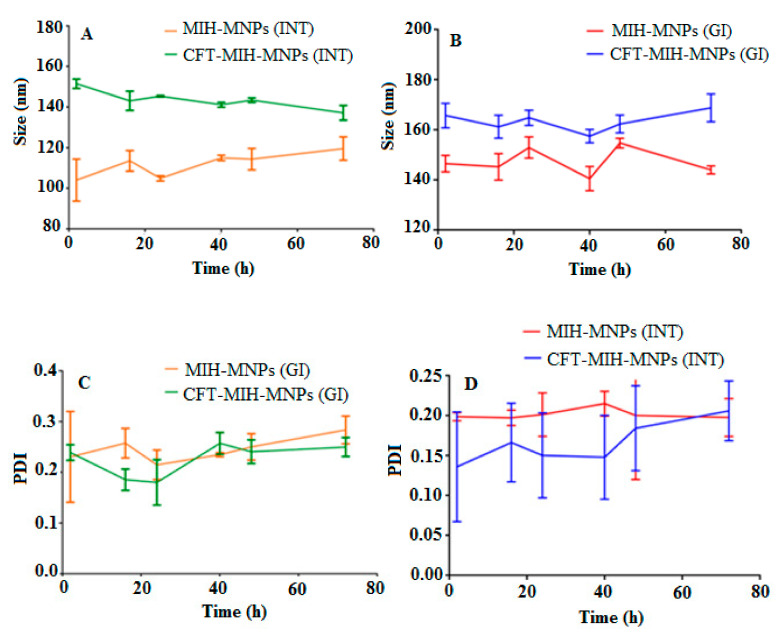
Hydrodynamic diameter of MIH-MNPs and CFT-MIH-MNPs under GIT conditions (**A**,**B**). PDI of MIH-MNPs and CFT-MIH-MNPs under GIT conditions (**C**,**D**).

**Figure 6 pharmaceutics-12-00492-f006:**
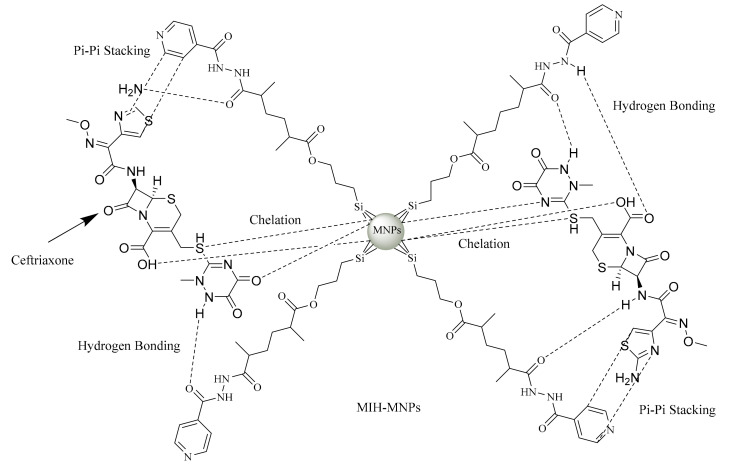
Aschematic diagram demonstrating the various possible interactionsbetweenceftriaxone (CFT) and MIH-MNPs.

**Figure 7 pharmaceutics-12-00492-f007:**
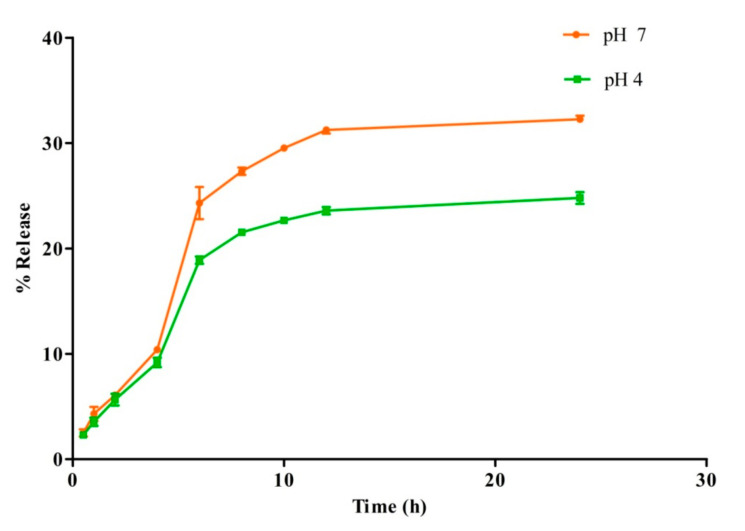
In vitro release profile of CFT-MIH-MNPs formulation at different pH values.

**Figure 8 pharmaceutics-12-00492-f008:**
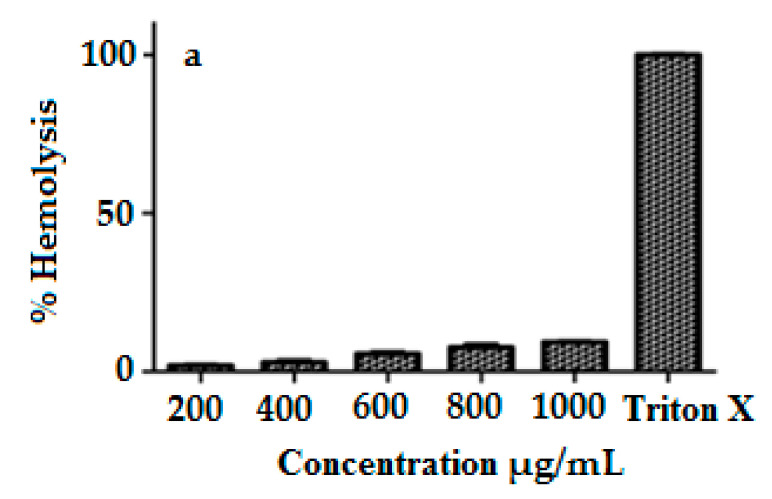
(**a**) Percent hemolysis of MIH-MP nano particle at different concentration. (**b**) In vitrocytotoxicity profile of MIH-MP nano particles against 3T3(NIH) and HeLa cell lines.

**Figure 9 pharmaceutics-12-00492-f009:**
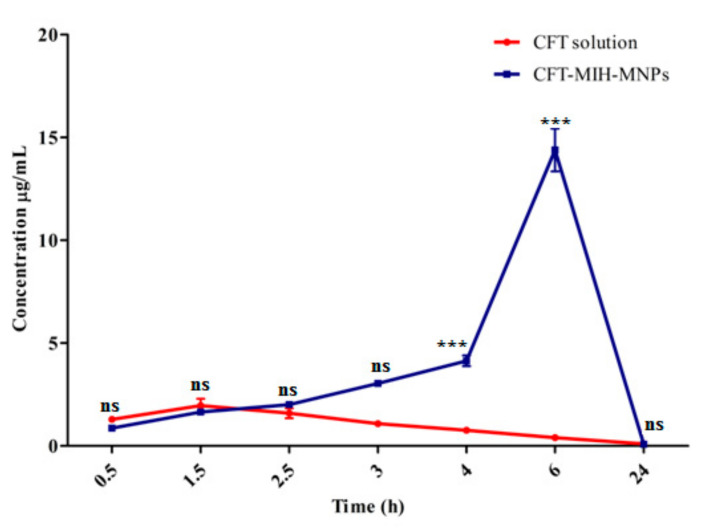
In vivo profile of CFT-MIH-MNPs against oral control at ^***^ extremelysignificant (*p* < 0.001), ^**^ highly significant (*p* < 0.01), ^*^ significant (*p* < 0.05), ^ns^ not significant (*p*> 0.05).

**Table 1 pharmaceutics-12-00492-t001:** Characteristics of CFT-MIH-MNPs formulation (average size, PDI and drug entrapment efficiency).

Sample	Average Size (nm)	Polydispersity Index (PDI)	Zeta Potential (mV)	Drug Entrapment Efficiency (%)
MPTES-MNPs	124 ± 3.4	0.232	–11.4 ± 1.3	-
CFT-MPTES-MNPs	143 ± 5.9	0.252	–18.3 ± 0.2	73.1 ± 2.4
MIH-MNPs	168 ± 7.3	0.237	–17.7 ± 0.4	-
CFT-MIH-MNPs	184 ± 2.7	0.265	–20.2 ± 0.4	79.4 ± 1.5

**Table 2 pharmaceutics-12-00492-t002:** In vivo oral pharmacokinetic parameters when administered CFT-MIH-MNPs loaded formulation, oral control and intravenous control.

Pharmacokinetic Parameters	CFT Solution	CFT-MIH-MNPs Formulation
Dose (mg/kg)	50.0 ± 0.3	25.0 ± 0.3
C_max_ (µg/mL)	2.0 ± 0.6	14.4 ± 1.8 ^***^
AUC_0-24_ (µg.h/mL)	16.1 ± 3.7	156.8 ± 0.3 ^***^
MRT (h)	8.4 ± 0.1	12.2 ± 0.2 ^***^
T_max_ (h)	1.5	6
Clearance (L/h)	3.9 ± 0.4	1.6 ± 0.2 ^***^
Volume distribution (L)	33.0 ± 1.94	10.9 ± 2.4 ^**^

^***^ Extremely significant (*p* < 0.001), ^**^ highly significant (*p* < 0.01), ^*^ significant (*p* < 0.05), ^ns^ not significant (*p* > 0.05)

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
