# Peer review of "Enhancement in Oral Absorption of Ceftriaxone by Highly Functionalized Magnetic Iron Oxide Nanoparticles"

_pharmaceutics, 2020, doi:10.3390/pharmaceutics12060492_

Round 1

Reviewer 1 Report

In this article, a N-methacryolisonicotinohydrazide (MIH)-functionalized MNPs were used as a drug delivery carrier of ceftriaxone (CFT) to protect its degradation by oral administration. This nano-drug system is synthesized by a simple route and is characterized by different techniques. In vitro drug release, in vitro cell toxicity of the carrier, and in vivo oral pharmacokinetics experiments show that this current nano-drug system is a promising novel strategy to enhance oral bioavailability of drugs like CFT.

Generally, in this research, the presented data support its conclusion. However, the following issues have to be addressed before considering acceptance.

  1. In the text, a brief introduction of the synthesis and preparation of the nano-drug system should be included. Section 3.1 and 3.2 are not about the synthesis and preparation, and they are part of the characterization.
  2. What is the point for 3.5 Determination of CFT in blood plasma? What are the data for this assay?
  3. In section 3.6 Drug Entrapment Efficiency. In order to explain a higher drug load in MIH-MNPs than in MPTES-MNPs, the authors speculate there is a stronger pi-pi stacking between CFT and the former. In my eyes, there is no big difference between entrapment efficiency of 79.4 ±1.5 % and 73.13 ±2.4 % considering the standard deviation. This result is over interpreted. Also, the structure of CFT should be presented to show pi-pi interaction.
  4. Please arrange figures according to their coming sequence in the text. E.g., Fig. 7 should be put before Fig. 6. Also for 3.7. In vitro drug release study, more discussion should be included since it shows the protection effect of nano-drug system in acidic environment in stomach.
  5. In 3.8 Hemocompatibility study. What reagent is used as a positive control? What does 10% hemolytic activity mean? Is it in the safety range? In 3.9 In vitro cytotoxicity. Why MIH-MNPs are more toxic to 3T3 than HeLa cells? You cannot say it is biocompatible when cell viability is only 73% after treatment.
  6. Another major issue for this article is its tedious writing. This text should be and can be largely shortened using more precise and neat language. Some background knowledge in result section, e.g., for 3.9 in vitro cytotoxicity, is not very necessary or should only be briefly mentioned. Some long paragraph should be divided into more small paragraphs (e.g., section 3.2). I suggestion omitting all numbers in the abstract.

Reviewer 2 Report

The Authors have used magnetic nanoparticles for loading the drug Ceftriaxone,  a drug that inhibits the synthesis of bacterial cell walls. Below are my concerns and suggested additions to improve the manuscript. 

  1. In the introduction, the authors discussed about disadvantages of administering Ceftriaxone. Can they possibly comment on the sub-lingual absorption of the drug - Similar to the other formulations generally used in medicine?
  2. In the sythesis of MIH-MNPs, the authors have clearly mentioned that the AIBN is used for radical polymerization. However, scheme in Fig 1 shoed only a single unit of MIH was added  to the particle surface.
  3. The drug Ceftriaxone is loaded through non-covalent interaction. However the zetapotential of drug-free NPs are negative. If the interaction between Ceftriaxone and MIH-MNPs are electrostatic, How do the authors explain the interaction of negatively charged drug to electrostatically interact with negatively charged particles?
  4. Were the zetapotential of MIH-MNPs and the CFT-MIH-MNPs measured at same pH? It is worth mentioning in the discussion.
  5. The work is important in its idea of using MNPs. The authors mush also recognize the previous efforts on oral delivery of Ceftriaxone and cite the works.(Pharmaceutical Research volume 30pages959–967(2013))
  6.  Also in conclusion, it is worth mentioning that the MNPs are great acoustic imaging agents and can be combined with techniques to tract drug release in-vivo with a citation. (Angew. Chem. Int. Ed. 202059, 4678.)

Reviewer 3 Report

Review report

Manuscript ID: 758674

Title: Enhancement in Oral Absorption of Ceftriaxone by Highly Functionalized Magnetic Iron Oxide Nanoparticles

Brief summary:

The manuscript aims at developing a biocompatible delivery system for the oral administration of ceftriaxone using functionalized iron oxide magnetic nanoparticles. The formulations were characterized in terms of hydrodynamic diameter, PDI, zeta potential, morphology, entrapment efficiency, biocompatibility using cell lines and hemocompatibility. Moreover, the pharmacokinetic parameters upon oral administration of the designed formulation were determined using an animal model, showing the ability of the formulation to increase the oral bioavailability of ceftriaxone in comparison with a commercial solution.

Broad comments:

The manuscript meets the scope of the journal as it intends to provide a novel nanomedicine to overcome the bioavailability issues of ceftriaxone (CFT) upon oral administration. The work provides the physicochemical characterization of the formulation together with the evaluation of its oral bioavailability upon administration in an animal model, which is the main strength of the manuscript.

After reading the manuscript, I believe that some points must be discussed in detail to clarify the rationale of the study and to include the results reported in the literature so far, as detailed below:

A. According to the authors: “CFT is poorly absorbed through the mucosal membrane of the intestine due to its two negatively ionized carboxyl groups” and CFT “belongs to class III drugs of biopharmaceutical classification system and suffers from lower membrane permeability”. In this context, the oral absorption of CFT would not be favored in the stomach due to their low pH content, favoring the protonation of the drug and increasing its membrane permeability? What are the reasons underlying the design of a nanoparticle that protects the drug from the gastric conditions? Are there evidences showing the degradation of the drug in these conditions?

B. Including an antibiotic as a functionalization moiety is, in my opinion, quite controversial. The rationale of including isoniazid as a functionalization moiety is not clear in the manuscript and I believe it should be very well explained and discussed, as it may have implications regarding the therapeutic efficacy of the formulation (will isoniazid be released in vivo and act together with CFT?). This rationale also raises questions regarding the antibiotic resistance, as it may lead to an increased exposure to antibiotics by the patients without significant therapeutic benefits.

C. Although the authors claim that the formulation is biocompatible based on in vitro studies, no results were provided regarding the biocompatibility of the formulation upon administration in the animal model used. This information would be valuable to improve the overall quality of the study, as well as evidences regarding the therapeutic efficacy of the designed formulation, which was completely disregarded in this study.

D. To improve the discussion of the work, the authors are encouraged to discuss their own results with the studies reported in the literature concerning nanomedicines for the delivery of CFT.

Specific comments:

Overall, the manuscript is well structured, but the English language may be enhanced to improve the clarity for the readers. Some suggestions to improve the clarity of the manuscript are described below:

A. Please check tables and text to include just the significant digits of each result. As an example, replace “184 ±2.7 nm” by “184 ±3 nm” in abstract.

B. Please correct minor typos throughout the introduction section, including:

Line 57: replace “can constitutea potential” by “can constitute a potential”

Line 60: replace “CFTis widely” by “CFT is widely”

Line 61: replace “CFT oral delivery isa real” by “CFT oral delivery is a real”

Line 63: replace “biopharmaceutical pharmaceutical system” by “biopharmaceutical classification system”

Line 67: replace “desirablefor oral delivery of CFT” by “desirable for oral delivery of CFT.”

Line 73: replace “the ability of Nps” by “the ability of NPs”

C. Check the sections: Materials and Methods, Results and Discussion and Conclusions for missing/extra black spaces and subject-verb agreement.

D. I suggest the figures to be numbered according to their appearance in the text (Fig. 7 is referenced before Fig. 6).

E. Please check figure captions: check the spelling of nanoparticles, as well as the correspondence between the caption and the figure (ex. Figure 7 caption includes the calibration curve, which is not displayed in the figure).

F. Line 364: replace Table 3 by Table 2.

G. I suggest the authors to improve the quality of Figure 4 and to discuss the morphology observed with CFT-MIH-MNPs in the manuscript. Indeed, the authors claim that the nanoparticles are spherical in the abstract, but is it supported by AFM data (Figure 4)?

Round 2

Reviewer 1 Report

Please use reagents' name instead of "control" in all figures. 

For section 3.5, I do not understand why development of a HPLC protocol is included here. Please state it clearly in the text. 

Author Response

Comment 1

For section 3.5, I do not understand why the development of HPLC protocol is included here. Please state clearly in the text.

Response

In order to quantify CFT in blood after administering our formulation in vivo, we developed and validated HPLC protocol of CFT in blood plasma as depicted in line 313-315. The values of accuracy and precision are presented in section 3.5 from line 316-320.

Comment 2

Please use reagent`s name instead of “control” in all figures.

Response

As shown in line 359 and 360 we have replaced the control with reagent`s name.